# Gender Affirmation through Correct Pronoun Usage: Development and Validation of the Transgender Women’s Importance of Pronouns (TW-IP) Scale

**DOI:** 10.3390/ijerph17249525

**Published:** 2020-12-19

**Authors:** Jae M. Sevelius, Deepalika Chakravarty, Samantha E. Dilworth, Greg Rebchook, Torsten B. Neilands

**Affiliations:** 1Center for AIDS Prevention Studies, Department of Medicine, University of California, San Francisco, CA 94158, USA; deepalika.chakravarty@ucsf.edu (D.C.); Samantha.Dilworth@ucsf.edu (S.E.D.); greg.rebchook@ucsf.edu (G.R.); torsten.neilands@ucsf.edu (T.B.N.); 2Center of Excellence for Transgender Health, Department of Medicine, University of California, San Francisco, CA 94158, USA

**Keywords:** scale development, scale validation, transgender women, gender affirmation, pronouns, HIV care, mental health

## Abstract

Social interactions where a person is addressed by their correct name and pronouns, consistent with their gender identity, are widely recognized as a basic and yet critical aspect of gender affirmation for transgender people. Informed by the Model of Gender Affirmation, we developed a self-report measure of the importance of social gender affirmation, the Transgender Women’s Importance of Pronouns (TW-IP) scale, which measures gender affirmation through the usage of correct pronoun by others. Data were from self-administered surveys in two independent samples of transgender women living with HIV in the US (N1 = 278; N2 = 369). Using exploratory factor analysis with data from Study 1 and confirmatory factor analysis with data from Study 2, we obtained a four-item scale with a single-factor structure and strong reliability (α = 0.95). We present evidence of TW-IP’s convergent and discriminant validity through its correlations with select mental health and HIV-related measures. Further, scores on TW-IP were linked in expected directions to several hypothesized mental health and HIV care outcomes, demonstrating its predictive validity. The resulting brief measure of importance of pronouns among transgender women shows strong psychometric properties. Validation evidence offers highly promising opportunities for use of the measure in clinical and research settings.

## 1. Introduction

Gender affirmation, in its broadest sense, refers to an interpersonal process whereby a person receives social recognition and support for their gender identity and expression [1]. The term ‘gender affirmation’ has become widely used to specifically describe the process whereby transgender individuals affirm their gender through social, legal, and/or medical pathways [2]. The construct of gender affirmation is situated within the larger Model of Gender Affirmation, which is informed by multiple theoretical frameworks, including intersectionality, objectification theory, and the identity threat model of stigma [1]. The Model of Gender Affirmation emphasizes that health disparities experienced by transgender people are rooted in intersectional stigma and underscores the importance of increasing access to all forms of gender affirmation as a means of reducing these health disparities. The Model of Gender Affirmation includes a range of hypothesized influences of gender affirmation on resilience, risk behavior, and engagement in healthcare.

Gender affirmation as a social process is inherently interactive, where one’s gender identity is affirmed through social interactions with others [1,3]. For example, social interactions where a person is addressed by their correct name and pronouns, consistent with their gender identity, are widely recognized as a basic yet critical aspect of gender affirmation. Social affirmation processes occur for cisgender people as well as transgender people; however, the term ‘affirmation’ is often associated with transgender and gender diverse people because they more commonly have experiences that are disaffirming [4]. These disaffirming experiences include being misgendered (e.g., addressed in a way that is inconsistent with one’s gender identity) through incorrect pronoun usage, for example, referring to a transgender woman as “he” when the woman’s “preferred” pronoun is “she”. However, transgender communities and their advocates have increasingly emphasized that pronoun use goes beyond “preference”, since “preference” implies that using the correct pronoun is simply “preferred” by the person and therefore optional [4]. Addressing someone by the wrong name or misgendering them through use of incorrect pronouns can feel disrespectful, harmful, and even unsafe to the person being misgendered, since misgendering results in marginalization and communicates that a person’s identity is not being seen or respected [5].

Positive experiences of social gender affirmation are critical to the health and well-being of transgender and gender diverse people. Greater social gender affirmation is associated with improved mental health and well-being among diverse groups of transgender and gender diverse adults and youth [3,6,7]. Social gender affirmation, when combined with healthcare empowerment, was associated with viral suppression among a large national sample of transgender women of color living with HIV in the United States [8]. Disaffirming experiences in healthcare settings, such as being misgendered using incorrect pronouns, can result from overt or implicit biases and lack of training among healthcare providers and staff, often leading to anticipation of stigma and the avoidance of healthcare among transgender people [5].

A primary obstacle to further exploration of social gender affirmation is a lack of psychometrically sound measures of the construct. We sought to develop a measure of the importance of social gender affirmation in alignment with the Model of Gender Affirmation and within the context of HIV-related health outcomes. We chose this context because extensive research has shown that active engagement in clinical care and high levels of adherence to antiretroviral treatment (ART) are essential for those with HIV to live longer, healthier lives. Transgender women living with HIV are impacted by disparities at every stage of the HIV care continuum [9,10].

The purposes of the current paper are to use secondary data from two large quantitative studies to (a) describe the development of the Transgender Women’s Importance of Pronouns (TW-IP) scale, a brief, self-report measure of gender affirmation through correct pronoun usage among transgender women living with HIV and (b) present evidence of the TW-IP’s relationship to variables that are hypothesized correlates as suggested by the Model of Gender Affirmation. These hypothesized correlates were informed by previous research and included variables related to hormone use, affect, and engagement in HIV care [6,8,11].

## 2. Methods

Item development: The potential items for this scale were developed as part of a qualitative study conducted to generate theory around the construct of gender affirmation. The recruitment methodology for that study has been described elsewhere, and included individual interviews with 22 transgender women of color to explore the impact of intersections of racism and transphobia on experiences of gender affirmation [1]. Interview transcripts were analyzed with Atlas.ti [12] using template analysis, a standard qualitative technique for identifying and organizing themes through the development of a coding template [13]. This technique is useful for analysis of qualitative data when some a priori themes are defined based on theory and/or the research questions of interest. In this case, the a priori themes were based on the theoretical framework of the Model of Gender Affirmation. Initial analysis led to the identification of thematic categories such as the need for gender affirmation and the importance of correct pronoun use, which were relevant to item development for the current study. The themes were then broken down into codes and modified as needed to reflect the language that the participants used to describe the constructs of interest (e.g., need for gender affirmation) to ensure that the coding template included culturally relevant language. This coding process yielded multiple dimensions of gender affirmation and generated an initial list of candidate items. Five of those items were relevant to the importance of correct pronoun use, which is the focus of this investigation.

These initial items were then tested using an iterative process involving two rounds of cognitive interviewing with 10 transgender women in the first round and 9 in the second round (total *N* = 19 unique participants). Cognitive interviewing is a technique used to improve the development of surveys by administering draft items while eliciting further information from the participants about their responses to the items [14]. Participants were asked to describe their understanding of each item and provide recommendations for improvement of the appropriateness, wording, and/or the ordering of items in the scale, as these issues arose. After the first round of cognitive interviews, the initial items were modified based on participants’ feedback, and then were further refined using the same technique in the second round of interviews. These final items were administered to participants in two larger quantitative studies as described below. The resulting items, including the five used for the current measure development, were administered to participants in two larger quantitative studies as described below.

Samples for the current analyses: Data used in the present analyses were from two independent samples of transgender women living with HIV, who participated in two distinct studies (hereafter referred to as Study 1 and Study 2).

Study 1—Study 1 was a randomized controlled trial of a theory-driven, population-specific intervention (“Healthy Divas”) conducted in San Francisco and Los Angeles, CA to improve engagement in care for transgender women living with HIV (Unique Protocol ID: NCT03081559). Grounded in models of Gender Affirmation and Health Care Empowerment, the intent of the study was to test the efficacy of an intervention designed to systematically intervene on complex barriers to optimal engagement in HIV care for transgender women living with HIV. Participants (*N* = 278) provided self-administered behavioral survey data at baseline and follow up visits. The current analysis utilizes the baseline data.

Study 2—As part of an initiative titled Enhancing Engagement and Retention in Quality HIV Care for Transgender Women of Color, nine demonstration sites across four urban centers in the US—Chicago, New York, Los Angeles and San Francisco Bay Area—developed and implemented innovative interventions to engage and retain transgender women of color living with HIV, in quality HIV care; an additional site provided technical assistance to the sites and evaluated the interventions [15]. Participants completed a self-administered survey at baseline and follow-up visits. The current analysis utilizes the baseline data. Since the communities of transgender women are small and tightly knit, many participants within California were common to both studies. Therefore, to ensure that the two samples consisted of non-overlapping participants for these analyses, we excluded from Study 2 all the participants recruited at the California study sites. We also excluded participants (*n* = 3) who did not respond to any of the items in the scale being developed, resulting in a final sample of 369 transgender women from Study 2.

### Screening and Recruitment

Study 1—Between November 2016 and October 2019, participants were recruited from community-based organizations, social networks, and local venues frequented by transgender women. Participants were eligible if they were 18 years or older; were assigned male sex at birth but identified as transgender female, female, or another transfeminine identity; were confirmed to be living with HIV (either via medical documentation or an HIV rapid test); and were fluent in English or Spanish. After obtaining informed consent, we collected baseline data via a self-administered survey using CASIC data collection software [16]. Survey questions included detailed information on participants’ health care, sexual risk, and their attitudes and behaviors around HIV care, ART medication adherence, stigma, and gender affirmation. To compensate for their time, participants received USD 40 for the baseline study visit, including eligibility screening. All participants gave their informed consent for inclusion before they participated in the study. The study was conducted in accordance with the Declaration of Helsinki. The study was approved by the Institutional Review Board at the University of California, San Francisco (15-17910), and the Western Institutional Review Board (20181370). 

Study 2—Between December 2013 and August 2016, participants were recruited using a variety of strategies including community outreach, networking, word-of-mouth, publicity materials, and referrals from clinics and other service providers. Participants were eligible if they were 16 years or older, assigned male sex at birth but identified as transgender or female, were living with HIV, and were fluent in English or Spanish. After obtaining informed consent, the participants completed a self-administered baseline survey in REDCap [17]. The survey covered topics such as their HIV care, health-related behaviors, and gender affirmation. To compensate for their time, participants received incentives between 25 and USD 50 across the sites. All participants gave their informed consent for inclusion before they participated in the study. The study was conducted in accordance with the Declaration of Helsinki. The study was approved by the local Institutional Review Board at each of the study sites, and the evaluation project was approved by the Institutional Review Board at the University of California, San Francisco (12-09622).

## 3. Measures

Candidate items for the new TW-IP scale: Both studies’ surveys contained these five questions: “How important is it to you that strangers call you ‘she’ when talking about you?”, “How important is it to you that family members call you ‘she’ when talking about you?”, “How important is it to you that your friends call you ‘she’ when talking about you?”, “How important is it to you that health care providers call you ‘she’ when talking about you?”, “How important is it to you to have a driver’s license or ID that says you are female?”. The five-point Likert-type response options were: Not at all important, Slightly important, Moderately important, Very important, Extremely important.

Sample characteristics: To characterize the two samples, we used the following data from the surveys: age, race-ethnicity, education, experiences of homelessness in the previous six months, engagement in sex work as a source of income in the previous six months, financial security (‘currently’ for Study 1, ‘in the previous six months’ for Study 2), and history of recent incarceration (‘in the previous twelve months’ in Study 1, ‘in the previous six months’ in Study 2).

Feminizing hormone therapy: Participants reported whether they had ever taken hormones (0 = No, 1 = Yes) and whether they were currently taking hormones (0 = No, 1 = Yes). 

HIV diagnosis and care: Participants reported whether they had been newly diagnosed with HIV in the previous six months, whether they had ever taken ART for the treatment of HIV (0 = No, 1 = Yes), and whether they were currently on ART (0 = No, 1 = Yes). 

Additionally, we used data on the following study-specific measures from the two studies: 

### 3.1. Study 1

Positive and negative affect: Current positive and negative affect were measured using the 10-item International Positive and Negative Affect Schedule Short Form (I-PANAS-SF); the two subscales each utilized five items. Cronbach’s alphas for internal consistency were 0.88 and 0.84 for Positive and Negative Affect, respectively. Items asked the participant to rate the frequency of each emotion; five unipolar response choices spanned from 1 (Never) to 5 (Always). Sample item: “Thinking about yourself and how you normally feel, to what extent do you generally feel upset?”

HIV stigma: Stigma around HIV status was investigated using a 12-item scale that asked the participant to indicate the frequency of each feeling using one of four unipolar response choices ranging from 0 (Not at all) to 3 (Often). Cronbach’s alpha was 0.92. Sample item: “I’ve felt ashamed of my HIV”.

HIV treatment knowledge: Participant’s understanding of HIV treatment and medication was measured using the 16-item HIV Treatment Knowledge Scale [18]. The response choices were 1 (True), 0 (False), and 7 (Don’t know). The sum of the correct responses was used for analysis; Cronbach’s alpha was 0.89. Sample item: “Over-the-counter herbal pills (e.g., St. John’s Wort) could make HIV medications less effective.”

Treatment expectancies—Ease: One subscale—anticipated ease of taking ART—of a 13-item Treatment Expectancies scale [19], was investigated using the mean of four items asking participants to rate their agreement with each statement; Cronbach’s alpha was 0.74. Five Likert response choices ranged from ‘strongly disagree’ to ‘strongly agree’. Sample item: “Taking HIV medications on schedule would be easy for me”.

Experiences of traumatic events due to crime: Various experiences of having been a target of criminal acts, ex. Robbery, mugging were investigated. Endorsed experiences were summed for analysis. Sample item: “Has anyone ever tried to take something directly from you by using force or the threat of force, such as a stick-up or mugging?”

Perfect adherence to ART: Adherence to ART was measured by two items: (1) the Visual Analog scale (VAS) [20], where the participant was asked to estimate their ART medication adherence in the past 30 days on a line from 0 to 100%, and (2) Safren’s [21] Likert item, “Thinking back over the past 30 days, rate your ability to take all your medications as prescribed.” Six bipolar response choices for the Safren item span from 0 (Very poor) to 5 (Excellent). Adherence to ART was categorized as perfect if VAS adherence was equal to 100 and the participant responded with a 5 (Excellent) on the Safren item.

Number of sexual partners: Among detailed questions about sex behavior in the previous 3 months, participants were asked “How many sexual partners have you had in the past 3 months?”. Questions in this section were open ended.

### 3.2. Study 2

*Depression:* Depression during the previous week was measured using a 10-item version of the Center for Epidemiological Studies Depression (CES-D) scale [22,23]. The Likert-type response options ranged from 0 (Rarely or none of the time (less than 1 day)) to 3 (Most or all of the time (5–7 days)). The scale scores were dichotomized for use in the predictive model (0: score less than 10, not depressed; 1: score of 10 or greater, depressed) [24]. Cronbach’s alpha for the interval-level depression score was 0.87. Sample item: ‘During the past week, I felt that everything I did was an effort.’

*Suppressed HIV viral load:* Participants were considered to be virally suppressed if their HIV viral load had been tested in the previous six months and they had an undetectable viral load at their last test. Thus, viral suppression was a binary variable (1: virally suppressed; 0: not virally suppressed).

## 4. Data Analyses

First, we used SAS software version 9.4 (SAS Institute, Inc., Cary, NC, USA) [25] to compute the proportions, means and standard deviations to obtain the descriptive statistics for both study samples.

Exploratory Factor Analysis (EFA): Next, we used the data from Study 1 on the five potential items for the new scale to perform the EFA. In the initial item screening stage, M*plus* version 8.4 (Muthén & Muthén: Los Angeles, CA, USA) [26] was employed to determine the number of factors to keep using the scree plot as well as the following fit statistics: root mean square error of approximation (RMSEA ≤ 0.06), the comparative fit index (CFI ≥ 0.95), and the standardized root mean square residual (SRMR < 0.08) [27]. Item factor loadings as well as the underlying theory dictated their retention in the final EFA model. The Hull method available in FACTOR 10 was used to check the number of factors retained in the initial screening [28].

Confirmatory Factor Analysis (CFA): Next, we used the data from Study 2 and the factor structure from the Study 1′s EFA to perform the CFA using M*plus* version 8.4. In the Study 2 sample, 6.77% participants (*n* = 25) had missing values on some but not all of the five candidate items for TW-IP. Therefore, while performing the CFA, we used multiple imputation within M*plus* with 250 imputed datasets [29]. We assessed the global model fit using the chi-square test of exact fit and the approximate fit using the following measures: the Root Mean Square Error of Approximation (RMSEA), Bentler’s Comparative Fit Index (CFI), and the Standardized Root Mean Square Residual (SRMR). Model-data fit is considered to be satisfactory if at least two of the following three conditions are met: RMSEA ≤ 0.06, CFI ≥ 0.95 and SRMR ≤ 0.08 [27]. Next, internal consistency reliability was estimated using Cronbach’s coefficient alpha using SAS PROC CORR (SAS Institute, Inc., Cary, NC, USA).

Convergent and discriminant validity: To evaluate these, we correlated the new scale with select measures of interest in the two studies. For convergent validity, we hypothesized that the new scale would be positively correlated with positive affect, negative affect, anticipated stigma for taking antiretroviral therapy (ART), and being on feminizing hormone therapy (ever and currently). On the other hand, for discriminant validity, we hypothesized that there would be no significant correlation of the new scale with knowledge of HIV treatment, anticipated ease of taking ART, experiences of traumatic events due to crime, and recent homelessness. These correlation analyses were conducted in M*plus* version 8.4.

Predictive validity: To evaluate this, we examined the bivariate association of the new scale with the following outcome variables from the two studies: being on ART (ever and currently), perfect adherence to ART, HIV viral suppression, depression, and the number of sexual partners. Since the number of sexual partners was a count variable, we performed negative binomial regression on it to obtain the incident rate ratio per unit change in the score of the new scale. The remaining outcomes were binary variables and we performed logistic regression on them to obtain the odds ratio per unit change in the score of the new scale. We hypothesized that higher scores on the new scale would be associated with greater odds (greater incident rate, in the case of number of sexual partners) of each of the outcomes. These analyses were conducted in SAS software V9.4.

## 5. Results

Participant characteristics: The descriptive characteristics of the two study samples are shown in Table 1. Participants in Study 1 were slightly older than those in Study 2 (mean age 43.5 years vs. 34 years in Study 2). Both studies were mostly composed of participants who were of Hispanic, Latina, or of Spanish origin (32.7% in Study 1 and 44.7% in Study 2) and participants who were non-Hispanic Black (45.3% in Study 1 and 51.2% in Study 2). The majority of participants in both studies had education levels that were grade 12 or lower, with similar levels of financial security and homelessness. Notably, a higher proportion of participants in Study 1 compared to those in Study 2 had ever taken ART (77.7% vs. 39.8%) and were currently on ART (68.7% vs. 37.4%).

Exploratory Factor Analysis of Study 1 data: The scree plot indicated one factor with an eigenvalue greater than 1.0, so the single-factor solution was chosen to model the items; two of the three fit statistics were achieved with one factor in M*plus* (CFI = 0.998, SRMR = 0.018) and the Hull method in FACTOR 10 also preferred one factor. All five items had factor loadings of 0.76 or greater and were retained (see Table 2).

Confirmatory Factor Analysis of Study 2 data: On performing a CFA of the five-item factor structure indicated previously by the EFA using the 250 imputed datasets, the test of exact model-fit rejected the null hypothesis of exact model-data fit (χ^2^(5) = 14.09, *p* = 0.015). However, the approximate model-data fit met the criteria of two out of the three fit statistics being within desirable bounds (RMSEA = 0.07, CFI = 1.00, SRMR = 0.007). We noted the extremely high correlation (.988) between two items (“How important is it to you that your friends call you “she” when talking about you?” and “How important is it to you that health care providers call you “she” when talking about you?”). We therefore re-ran the CFA by dropping the first of these two items with the reasoning that one’s friends are usually most likely to use the correct pronouns and, as a result, retaining this item is unlikely to provide a high degree of additional useful information. During this second run, the fit statistics improved appreciably and indicated satisfactory exact and approximate model fit (χ^2^(2) = 1.43, *p* = 0.489; RMSEA = 0.008, CFI = 1.00, SRMR = 0.001). The final CFA factor loadings for the four items are presented in Table 2; the final scale is presented in the Appendix A. The reliability was high for this four-item scale (α = 0.95).

Convergent and discriminant validity using Study 1 and Study 2 data: As hypothesized, the new scale was positively and significantly (*p* < 0.05) correlated with positive affect, negative affect, anticipated stigma for taking ART, and being on feminizing hormone therapy both ever and currently (Table 3). Further, as hypothesized, the new scale was not significantly correlated with knowledge of HIV treatment, anticipated ease of taking ART, experiences of traumatic events due to crime, and recent homelessness.

*Predictive validity using Study 1 and Study 2 data:* The results from the logistic regression of the new scale on select outcome variables showed a statistically significant positive association in each case (Table 4). Specifically, higher scores on TW-IP were associated with higher odds of ever being on ART (OR = 1.28, *p* = 0.04), currently being on ART (OR = 1.25, *p* = 0.04), perfect adherence to ART (OR = 2.10, *p* = 0.02), HIV viral suppression (OR = 1.34, *p* = 0.003) and depression (OR = 1.26, *p* = 0.01). Similarly, the negative binomial regression demonstrated that higher scores on TW-IP were associated with a higher incident rate for the number of sexual partners (IRR = 2.27, *p* = 0.002).

## 6. Discussion

The results of the factor analyses and convergent validity analysis suggest that a single-factor structure of TW-IP fit the data well in the two research samples, and that importance of pronouns can be measured with a parsimonious four-item measure (see Appendix A for the final scale). We began with a five-item measure and found that the scale performed better with the “friends” item dropped, suggesting that the importance of correct pronoun usage by strangers, family, and healthcare providers is sufficient to measure the TW-IP construct. This finding may be because interactions with one’s friends are less likely to be disaffirming. We also included one item related to the importance of having a legal document (e.g., driver’s license) that reflects one’s correct gender marker. While this item overlaps with the construct of legal gender affirmation, we found that it aligned conceptually with the pronoun use items designed to measure social gender affirmation. This alignment may be because legal documents, especially those such as a driver’s license, are often used to identify oneself to others and must be presented in various social situations (i.e., airports, entrances to venues), thereby having a significant impact on whether a person’s gender is socially affirmed in those interactions.

As predicted by the Model of Gender Affirmation, scores on the TW-IP were linked in expected directions to hormone use and positive affect. Gender affirmation for transgender people is often associated with access to transition-related medical care, such as hormones and surgery, so the positive correlation between scores on the TW-IP and hormone use was not surprising. It is clear though, that social gender affirmation and medical gender affirmation are distinct, reflecting the fact that not all transgender women who deem it important to be addressed by the correct pronouns use hormones and vice versa. These findings highlight how transgender women who deem it important to be addressed by correct pronouns are engaging in self-affirmation and self-advocating for being socially affirmed, which are important indicators of self-determination and empowerment.

Scores on the TW-IP were associated with negative affect and expected stigma from taking ART. While we typically think of positive and negative affect as inversely related, in fact they often vary independently [30]. Transgender women with high scores on the TW-IP may be particularly susceptible to the negative impacts of microaggressions related to gender disaffirming experiences of being addressed by incorrect pronouns, thus resulting in parallel increases in negative affect and anticipated stigma. Exploration of several factors that were not hypothesized by the Model of Gender Affirmation to be associated with scores on the TW-IP included HIV treatment knowledge, expected ease of taking ART, experiences of traumatic events due to crime, and experiences of homelessness in the prior 6 months. The lack of association between TW-IP scores and these measures contribute to the establishment of discriminant validity of the TW-IP scale. Further, the TW-IP scale demonstrated important associations with hypothesized health outcomes and HIV-related risk behaviors, including ever having taken ART, currently taking ART, perfect adherence to ART, being virally suppressed, and number of sex partners. 

The Model of Gender Affirmation explicitly acknowledges that the experiences, identities, and preferences of transgender and gender diverse people are not homogeneous. Several studies have shown that correct pronoun use is critical to successful engagement and retention of transgender people in health care and that incorrect pronoun use can result in healthcare avoidance [31,32,33,34]. However, to the best of our knowledge, no quantitative measure exists to capture the importance of correct pronoun usage among transgender people, and further, no quantitative studies have explicitly looked at the impact of pronoun use on social gender affirmation or HIV-related health outcomes, which speaks to the novelty of our findings and the unique contribution of this measure. Transgender and gender diverse people use many different terms and pronouns to describe their identities, and these can evolve over one’s lifetime [35]. Further, the importance of the usage of particular pronouns varies among transgender people. For many, correct pronoun usage is a critical, respectful, and affirming form of communication that builds trust in settings such as healthcare [31]. Asserting one’s gender identity, correct name and pronoun, and insisting on the importance of having these respected by others may represent an important form of self-actualization and empowerment among transgender people, as well as a pathway to increase social gender affirmation [33,36].

### 6.1. Strengths and Limitations

This study was conducted among community- and clinic-based samples of transgender women living with HIV. Because these were not probability samples, our results may not generalize to all transgender women or transgender women not living with HIV. While development and initial validation of the TW-IP took place with transgender women living with HIV, the majority of whom were women of color, the scale is designed to be applicable across populations of transgender women regardless of HIV status or race.

The relatively moderate sample sizes and limited variability of age, race, and ethnicity preclude specific analysis of subgroups. However, in terms of population-specific research, these are two of the largest samples of transgender women living with HIV to date. Further, these samples were geographically diverse as they came from different regions of the US (West, East, and Midwest) and the majority of participants were women of color. Finally, an additional strength of this study is that our results were validated using two independent samples and we achieved consistent results across both samples.

### 6.2. Future Directions

The TW-IP was developed and tested with transgender women who were assigned ‘male’ sex at birth but identified as women, transgender women, or had another transfeminine identity. It is possible that some of our participants did not prefer “she” pronouns, as the diversity of pronoun preferences among transgender and gender diverse people has grown exponentially in the past few years. Future research should investigate how the importance of pronouns functions for other gender diverse people, such as transgender men and non-binary individuals. An adapted version of the TW-IP scale could also be explored with people who prefer non-standard pronouns, such as “ze/zir/zirs”, those who prefer no pronouns at all, and those who prefer that others vary the pronouns they use when addressing them. Future studies should also collect sufficiently large samples to permit testing the invariance of the measure’s performance across different demographic (e.g., age, race/ethnicity, education), psychographic (e.g., prefer non-standard pronouns vs. standard pronouns), and health status (e.g., HIV status) variables.

Future research should examine longitudinal patterns of TW-IP scores and predictive associations with engagement in healthcare, treatment adherence and persistence, and clinical outcomes, including health outcomes among transgender women not living with HIV. Among such outcomes are health-related quality of life, satisfaction with care, stigma resilience, and mental health. The availability of a psychometrically sound measure offers opportunities to investigate potential gender affirmation-related drivers of disparities in health-related outcomes experienced by transgender women. Additional research with sufficiently large and longitudinal samples is needed to evaluate the test–retest reliability of the measure and to test hypothesized determinants and consequences of TW-IP scores through associations with other constructs such as gender identity, socioeconomic status, and positive affect over time.

## 7. Conclusions

In summary, the TW-IP scale is a novel, brief measure of importance of pronouns among transgender women with strong psychometric properties, and preliminary validation evidence offers highly promising opportunities for use of the measure in clinical and research settings. The TW-IP scale has the potential to inform research that explores the impact of correct pronoun use as a critical form of gender affirmation on engagement in healthcare and other health outcomes among transgender women. That research can in turn serve to inform gender-affirming clinical practices that seek to address health disparities among transgender women, including HIV, mental health, and other health conditions that require active participation in health care.

## Figures and Tables

**Table 1 ijerph-17-09525-t001:** Descriptive characteristics of participants in Study 1 (*N* = 278) and Study 2 (*N* = 369).

Characteristic	Study 1		Study 2	
Age in years—mean (std. dev)	43.5	(10.7)	34.0	(10.8)
	*n*	(%)	*n*	(%)
Race-Ethnicity				
Hispanic, Latina, or of Spanish origin	91	(32.7)	165	(44.7)
Black, non-Hispanic	126	(45.3)	189	(51.2)
White, non-Hispanic	19	(6.8)	-	-
Asian or Pacific Islander, non-Hispanic	8	(2.9)	1	(0.3)
Additional, non-Hispanic	3	(1.1)	1	(0.3)
Multiracial, non-Hispanic	30	(10.8)	5	(1.4)
No response	1	(0.4)	8	(2.2)
Education				
Less than grade 12	78	(28.1)	122	(33.1)
Grade 12	109	(39.2)	144	(39.0)
Some college or higher	91	(32.8)	86	(23.3)
No response	0	(0)	17	(4.6)
Financially secure ^1^	49	(17.6)	88	(23.9)
Experienced homelessness in previous 6 months	114	(41.0)	163	(44.2)
Sex work as a source of income in previous 6 months	50	(18.0)	133	(36.0)
Incarcerated recently ^2^	75	(27.0)	24	(6.50)
Currently taking hormones	187	(67.3)	173	(46.9)
Newly diagnosed with HIV in previous 6 months	30	(10.8)	97	(26.1)
Ever taken ART	216	(77.7)	147	(39.8)
Currently on ART	191	(68.7)	138	(37.4)

^1^—Study 1: ‘Currently’, Study 2: ‘in the previous 6 months’; ^2^—Study 1: ‘in the previous 12 months’, Study 2: ‘in the previous 6 months’; ART—Antiretroviral therapy for HIV.

**Table 2 ijerph-17-09525-t002:** Standardized factor loadings from factor analyses.

Question Text	EFA Loading (Study 1)	CFA Loading (Study 2)	95% Confidence Interval of CFA Loading
	(*N* = 278)	(*N* = 369)	
How important is it to you…			
… that strangers call you “she” when talking about you?	0.870	0.992	(0.985–0.998)
… that family members call you “she” when talking about you?	0.851	0.948	(0.933–0.963)
… that your friends call you “she” when talking about you?	0.932		
… that health care providers call you “she” when talking about you?	0.915	0.966	(0.951–0.978)
… to have a driver’s license or ID that says you are female?	0.759	0.910	(0.885–0.934)

Notes: EFA factor loadings were estimated using FACTOR 10; CFA factor loadings and confidence intervals were estimated using M*plus* 8.4.

**Table 3 ijerph-17-09525-t003:** Correlations of Transgender Women’s Importance of Pronouns (TW-IP) with select study measures.

Measures	Data Source	Correlation	95% Confidence Interval	*p*-Value
Convergent Validity				
Positive affect	Study 1	0.227	(0.115–0.338)	<0.001
Negative affect	Study 1	0.154	(0.039–0.269)	0.008
Anticipated stigma for taking ART	Study 1	0.316	(0.146–0.486)	<0.001
Ever on hormones	Study 2	0.208	(0.108–0.307)	<0.001
Currently on hormones	Study 2	0.17	(0.069– 0.271)	0.001
Discriminant Validity				
HIV treatment knowledge	Study 1	0.054	(−0.063–0.171)	0.369
Anticipated ease of taking ART	Study 1	0.014	(−0.176–0.203)	0.889
Experiences of traumatic events due to crime	Study 1	0.026	(−0.091–0.144)	0.66
Experienced homelessness in past 6 months	Study 2	0.059	(−0.050–0.169)	0.288

TW-IP—Scores on the Transgender Women’s Importance of Pronouns scale; ART—Antiretroviral therapy for HIV; Sample size: 278 (Study 1), 369 (Study 2). Correlations estimated using full information maximum likelihood (FIML) in M*plus* 8.4.

**Table 4 ijerph-17-09525-t004:** Bivariate associations of TW-IP with select outcomes.

Outcome	Data Source	Odds Ratio	95% Confidence Interval	*p*-Value
Ever on ART	Study 1	1.278	(1.013–1.616)	0.039
Currently on ART	Study 1	1.249	(1.010–1.546)	0.041
Perfect adherence to ART ^1^	Study 1	2.104	(1.141–3.880)	0.017
Suppressed HIV viral load	Study 2	1.339	(1.107–1.619)	0.003
Depressed ^2^	Study 2	1.260	(1.056–1.504)	0.011
		**Incident Rate Ratio**	**95% Confidence Interval**	***p*-Value**
No. of sexual partners	Study 1	2.268	(1.088–1.478)	0.002

TW-IP—Scores on the Transgender Women’s Importance of Pronouns scale; ART—Antiretroviral therapy for HIV; Sample size: 278 (Study 1), 369 (Study 2). ^1^—Calculated for participants on ART (*N* = 191); ^2^—Calculated for participants with non-missing scores for depression (*N* = 343).

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
