# Peer review of "Gender Affirmation through Correct Pronoun Usage: Development and Validation of the Transgender Women’s Importance of Pronouns (TW-IP) Scale"

_ijerph, 2020, doi:10.3390/ijerph17249525_

Round 1
Reviewer 1 Report
This study outlines the development and validation of the Transgender Women’s Importance of Pronouns scale through the use of qualitative interviews, cognitive interviews and two distinct samples of transgender women.
The paper is generally clear and well written, and the rationale for the study and context used is well-explained. The descriptions of study 1 and study 2 were generally clear and easy to follow as well. Below are some comments regarding some specific parts of the manuscript:
p2, Line 45 “however, the term ‘affirmation’ is often associated with transgender and gender diverse people because they more commonly have experiences that are disaffirming“ - A citation here would be beneficial.
p2, Line 54 “Addressing someone by the wrong name or misgendering them through use of incorrect pronouns can feel disrespectful, harmful, and even unsafe to the person being misgendered” – I am unsure why these disaffirming experiences might be considered “unsafe” for the person being misgendered. If the following paragraph is meant to address this statement, the link should be made clearer.
p2, Line 78 “…with 22 transgender women of color…” – It is unclear why only women of color were recruited for the qualitative interviews since there was no distinction between transgender women of color versus white transgender women up to this point.
p2, Line 89 “This coding process generated the initial list of candidate items for the scale.” – It would be beneficial to state how many items were included in the initial list. It is good if the authors are able to list the initial items in a table or an appendix
p2, Line 91 “…with 10 transgender women in the first round and 9 in the second round…” – It is currently unclear if the two samples are independent. It would be beneficial to explicitly mention if the samples were independent samples consisting of a total of 19 unique transgender women.
p3 Line 103 "Study 1 was a randomized controlled trial of a theory-driven..." I think it will be good for the authors to cite the randomized controlled trial if it is already published
pp. 4-5 It will important for the authors to report the internal consistency of the measures (e.g., I-PANAS-SF, HIV stigma, CES-D, etc).
p. 5 It is unclear why EFA was conducted only on the five items. It is not clear why the authors did not conduct EFA on the initial items?
p. 8 Measurement invariance test should be conducted to examine whether the same construct is being measured across demographic factors such as age groups and education level.
Reviewer 2 Report
I enjoyed reading your work. Here are some of my concerns:
1. Response biases are not mentioned. For example, participation bias, participants who immediately completed the questionnaire were perhaps more interested in the topic. How do you think this might influence the responses?
2. Is the sample size representative for each group? With what level of confidence? If not, how do you limit the sample sizes to your conclusions?
3. In the discussion I recommend making a comparison between similar studies. What findings have been found in other investigations on gender affirmation through the correct use of pronouns?
Reviewer 3 Report
The manuscript submitted for review examines a topic of great relevance in the field of psychology. The topic is very important and the conceptual analysis made in the text is quite deep. The literature consulted is quite current and the analyzed sample isn´t very large. I would like to thank the efforts by the authors of the manuscript and congratulate them on the work. Overall, the writing is clear, the goals are well described, well-considered introduction and the results properly made and presented. Therefore, the manuscript brings significant knowledge of the scientific literature so and still covers existing gaps in the field. On a formal level, the manuscript is perfectly structured and the references comply with the rules. The work is ambitious and the results confirm the most of the hypotheses and the relevance and potential of the work is therefore recognized, but this Reviewer considers that several changes are needed to the manuscript is publishable.
Personally, I consider that the hypotheses of the study should be explained in more detail taking into account previous research. The Method section should include a slightly more detailed explanation of the instruments, including the psychometric properties. In addition, it must be described if they have informed consent and comply with ethical standards.
Overall it's a good work that could be improved adding greater conceptual clarity in the ideas presented. Personally I would focus on describe and explain what is the novelty of the work compared to other published studies on same topic.
It assumes a good work of study conforms to the objectives and establishes a good starting point for further research on the topic and its practical implications.
Round 2
Reviewer 1 Report
The authors have sufficiently addressed all my comments. I appreciate the authors' effort and hard work. Well done!